# Metabolic Status of Lean and Obese Zucker Rats Based on Untargeted and Targeted Metabolomics Analysis of Serum

**DOI:** 10.3390/biomedicines10010153

**Published:** 2022-01-12

**Authors:** Stepan Melnyk, Reza Hakkak

**Affiliations:** 1Arkansas Children’s Research Institute, 13 Children’s Way, Little Rock, AR 72202, USA; MelnykSB@archildrens.org; 2Department of Dietetics and Nutrition, University of Arkansas for Medical Sciences, 4301 W. Markham St., Little Rock, AR 72205, USA; 3Department of Pediatrics, University of Arkansas for Medical Sciences, 4301 W. Markham St., Little Rock, AR 72205, USA

**Keywords:** obesity, serum metabolomics

## Abstract

Obesity is growing worldwide epidemic. Animal models can provide some clues about the etiology, development, prevention, and treatment of obesity. We examined and compared serum metabolites between seven lean (L) and seven obese (O) female Zucker rats to investigate the individual serum metabolic profile. A combination of HPLC-UV, HPLC-ECD, and LC-MS revealed more than 400 peaks. The 50 highest quality peaks were selected as the focus of our study. Untargeted metabolomics analysis showed significantly higher mean peak heights for 20 peaks in L rats, generally distributed randomly, except for a cluster (peaks 44–50) where L showed stable dominancy over O. Only eight peaks were significantly higher in O rats. Peak height ratios between pairs of L and O rats were significantly higher at 199 positions in L rats and at 123 positions in O rats. Targeted metabolomics analysis showed significantly higher levels of methionine, cysteine, tryptophan, kynurenic acid, and cysteine/cystine ratio in L rats and significantly higher levels of cystine and tyrosine in O rats. These results contribute to a better understanding of systemic metabolic perturbations in the obese Zucker rat model, emphasizing the value of both whole metabolome and individual metabolic profiles in the design and interpretation of studies using animal models.

## 1. Introduction

Obesity is an intensely growing worldwide epidemic. According to the World Health Organization (WHO), the global number of obese people has tripled between 1975 and the present day. In 2016, approximately 1.9 billion adults were overweight, including over 650 million who were obese [1]. The latest report from the Center for Disease Control (CDC) indicates that the US obesity rate jumped from 30.5% to 42.4% between 1997/1998 and 2017/2018, and the rate of severe obesity increased from 4.7% to 9.2% during the same time [2,3]. Obesity-related health issues such as heart disease, stroke, type 2 diabetes, liver diseases, and certain types of cancer are the leading causes of preventable and premature death [4,5]. In 2008, obesity-related medical care costs in the US were estimated to be USD 147 billion [6,7]. The major contributor to the epidemiological explosion of obesity in Westernized countries is the complexity of the problem, which includes social, genetic, and epigenetic influences. Individual variability contributes to the heterogeneity of this problem and requires a personalized solution [8].

Despite intense research efforts in the past several decades, a decisive, acceptable, and unified classification of obesity or effective strategies for obesity prevention is still not available [9]. Current research efforts should focus on studying the basis of obesity and applying new technical advantages and knowledge.

A variety of animal models has been used to replicate human obesity. Generally, these animal models can be divided into genetic (monogenic, polygenic, transgenic) manipulations and non-genetic (dietary, surgical, small, and large animals) [10]. Though animal models remain indispensable for investigating the causes, development, treatment, and prevention of obesity, they cannot fully replicate human obesity and can only provide clues into the etiological factors of obesity, development of disease, preventive procedures, and treatment options.

The development of obesity in different animal models can be influenced by several factors, including biological, psychosocial, and environmental factors, which must be considered in experimental design, data analysis, and data interpretation. Despite the known limitations of animal obesity models, they have significant advantages, such as known genetic manipulations, shorter lifespan, cost-effectiveness, standard diets, ability to be raised in colonies, and ability to represent human diseases, including obesity [11].

How genetic manipulations affect metabolic pathways in animal models and the degrees of the effects are important and need to be clarified. Metabolomics could be an appropriate and helpful option in such investigations. Together with numerous other *-omics* branches, metabolomics has had tremendous development and use in research within the past two decades.

The metabolome can be defined as the total amount of small molecular weight original biomolecules and their metabolites that are present in any biological system. Their detection and profiling (untargeted metabolomics), identification, and quantification (targeted metabolomics) are capable of providing objective and precise information about an individual’s metabolic phenotype and a metabolic snapshot of disease conditions. Such analytical sequences will build the basis for the discovery of biomarkers of pathological conditions, including obesity [12,13].

Untargeted metabolomics is an approach capable of utilizing a variety of chromatographic techniques, including “traditional” HPLC with different types of detection (UV, ECD) and liquid chromatography-mass spectroscopy (LC-MS, LC-MS/MS). Metabolomics analysis is capable of generating a high volume of information in the form of peaks related to the quantity of all detectable metabolites in a sample, creating an individual metabolic profile. These peaks can be collected, systemized, statistically analyzed, and interpreted, making an experimental group profile. Data can also be collected and processed as a case/control pair comparison to identify their differences. Despite some protocol limitations, untargeted metabolomics analytical workflow can be used as an initial and sequential process to better the understanding and interpretation of biological processes. This technique has a unique potential for the early or discovery stage of a study of pathological processes and is capable of unveiling new approaches in the research or expanding existing knowledge on the object of the research [14].

For the past several years, our laboratory has used the Zucker rat model to investigate the effects of obesity and different diets on the development of breast cancer and non-alcoholic fatty liver disease [15].

Zucker rats are the most widely used rat model for genetic obesity and have been used to study noninsulin-dependent diabetes mellitus and other diseases [16,17,18,19]. Animals homozygous for the fa allele are notably obese by 3 to 5 weeks of age and by 14 weeks of age, their body composition is more than 40% lipid [20]. A number of investigators have used this model to study the development, etiology, associated pathogenesis, possible treatment, and putative mechanisms of severe obesity [21]. Obese Zucker rats develop hyperinsulinemia and insulin resistance prior to the development of obesity-associated, non-insulin-dependent diabetes mellitus in a manner similar to that in humans [22]. Lean Zucker rats, by contrast, exhibit normal metabolic function and are considered ideal controls [23]. For the past several years, we used this model to investigate the effects of both short- and long-term soy diet on liver steatosis. We reported that obesity increases liver steatosis and short- and long-term soy protein diet can protect liver steatosis [24,25]. In addition, we have used the obese Zucker rat model to study the effects of obesity on breast cancer development and reported that obesity increases mammary tumor development [26,27].

In the present study, we aimed to investigate and compare serum metabolic data from lean and obese Zucker rats. Creating individual and group metabolic profiles will be a helpful tool for a better understanding of the specificity of this model and for widening the base for future manipulations. This knowledge will contribute to the clarification of some etiological, pathogenic, and prognostic aspects of obesity. For the completion of this study, we used a combination of HPLC-UV, HPLC-ECD, and LC-MS analytical platforms and were able to collect more than 400 peaks with different levels of intensity. We selected the 50 most intense and clean peaks for evaluation in this study.

## 2. Materials and Methods

### 2.1. Experimental Design

The animal protocols used in this study were approved by the Institutional Animal Care and Use Committee at the University of Arkansas for Medical Sciences.

Fourteen, 5-week-old female Zucker rats (7 lean and 7 obese fa/fa) were purchased from Harlan Industries (Indianapolis, IN, USA). We have shown that both obese male and female rats will develop obesity and liver steatosis at the same rates and that there is no difference on between both sexes [2,25]. Rats were housed one per cage and had ad libitum access to water and a semi-purified diet similar to AIN-93G diet (Envigo, Madison, WI, USA) for 18 weeks. Rats were weighed twice per week. All rats were sacrificed at the end of the experiment, and sera were collected and kept at −80 °C prior to analysis.

### 2.2. Sample Preparation for Measurement of Serum Metabolites

For the determination of non-protein bonded metabolites, proteins were precipitated by the addition of 100 μL ice-cold 10% meta-phosphoric acid to 100 μL of serum. The sample was then incubated for 10  min on ice, followed by centrifugation at 18,000×  *g* for 15 min at 4 °C. The resultant supernatant was filtered through a 0.2  μm nylon filter, and a 20 μL aliquot was analyzed via HPLC-ECD and HPLC-UV.

For determination of non-protein bonded metabolites, 50 μL of serum was mixed with 300 μL of HPLC-grade methanol followed by centrifugation t 18,000× *g* for 15 min at 4 °C. Supernatants were dried under nitrogen flow and then suspended in 50 μL of 50% methanol/0.2% formic acid. A 5 μL sample was then analyzed via LC-MS.

### 2.3. HPLC and LC-MS Methods

#### 2.3.1. HPLC with Coulometric Electrochemical Detection (ECD)

All methodological details for metabolite elution, detection, and analysis have been described previously [28,29] Aliquots of serum extract (20 μL) were analyzed using HPLC with an ESA solvent delivery system (model 580, ESA Inc., Chelmsford, MA, USA), a Phenomenex Capcell Pak reverse phase C_18_ column (4.6 × 150 mm, 3 μm particle size) (Phenomenex Inc., Torrance, CA, USA), an ESA autosampler (model 507E), dual analytical cell (model 5010), and guard cell (model 5020). The mobile phase was composed of 50 mM sodium phosphate, 1.0 mM octanesulfonic acid, and 2% acetonitrile (*v*/*v*) at pH 2.7, with an isocratic flow rate of 1 mL/min, with a column compartment thermostat temperature of 25 °C. The concentration of serum metabolites was calculated from peak areas of the sample and standard calibration curves using HPLC software.

#### 2.3.2. HPLC with UV Detection

HPLC-UV was performed on aliquots of serum extract (20 μL) with a Thermo Scientific UltiMate 3000 system (Thermo Fisher Scientific Inc., Waltham, MA, USA) equipped with a pump, autosampler, column compartment, and RS variable wavelength UV diode array detector. The chromatographic separation of metabolites was performed using a Raptor AR C_18_ Column (150 × 4.6 mm, 2.7 μm particle size) (Restek Co., Bellefonte, PA, USA) under isocratic elution. The mobile phase contained 0.05 mol/L KH_2_PO_4_ and 10% acetonitrile (*v*/*v*). The UV detector was set at variable wavelengths (220, 240, 260, 280 nm). The concentrations of plasma metabolites were calculated from peak areas and standard calibration curves using HPLC software.

#### 2.3.3. LC-MS Detection

LC-MS was performed with a Thermo Fisher Scientific LTQ XL mass spectrometer coupled with a Thermo Scientific UltiMate 3000 HPLC system. The MS detector was set with a +3.5 kV spray voltage, a 275 °C capillary temperature, a 300 °C heater temperature, a sheath gas flow of 25 L/min, auxiliary gas flow of 10 L/min, and a mass scan range of 80–900 *m*/*z*.

The gradient mobile phases were (A) 10 mM ammonium formate in 95:5 acetonitrile/water + 0.1% formic acid and (B) 10 mM ammonium formate in 50:50 acetonitrile/water + 0.1% formic acid for 30 min at a flow rate of 300 µL/min. Aliquots of serum extract (5 μL) were injected into a Thermo Fisher Scientific Accurore C_18_ Column (100 × 2.1 mm, 2.6 μm particle size), with a column temperature of 40 °C. MS data acquisition was performed using Thermo Fisher Scientific Xcalibur™ software v 2.2 SP1.48. Metabolites were determined through comparison of the ion features in the experimental samples to those of control samples based on retention time and molecular weight (*m*/*z*) using Thermo-Fisher Scientific Mass Frontier 7.0 software [30,31].

### 2.4. Statistical Analysis

The statistical analyses were performed using ANOVA and SigmaPlot 13.0 software (Systat Software, Inc., San Jose, CA, USA). The results are presented as mean ± SD, and Student’s two-tailed *t* test was used for independent comparisons of the heights of common peaks between the control (L) and obese (O) groups of Zucker rats. A value of *p* < 0.05 was considered a significant difference.

## 3. Results

### 3.1. Body Weight

The mean body weights in grams (mean ± SD) at the beginning of the experiment were 97.8 ± 6.5 for lean (L) and 154.8 ± 13.3 for obese (O) rats, and at the end of the experiment, the body weights were 270.8 ± 25.0 for L and 590.2 ± 41.0 for O rats.

### 3.2. Untargeted Metabolomics Data

From the collected untargeted metabolomics data, we identified over 400 high-quality peaks (clean peak on the chromatogram) with low noise to peak ratios (1:10 noise to peak ratio minimum). From these, we selected the 50 highest-quality peaks. We compared identical peaks between L and O rats. Our assumption was that every peak represented only one metabolite.

Serum of L rats contained a significantly higher average height in 20 peaks compared to O rats (Figure 1A). The highest peaks in the L group were distributed randomly across the chart until peaks 44 through 50, among which L rats showed stable dominancy over O rats. In contrast, O rats developed significantly higher average peak intensity in only 8 peaks that were distributed randomly across the chart. There was no difference in average peak intensity in 22 positions, which included three peaks in which the differences were on the margin of significance (0.1 > *p* > 0.05).

After comparing average peak heights between L and O groups, we analyzed individual peak height ratios (L/O) between seven pairs of L and O rats for our 50 peaks of interest; this resulted in 350 data points (Figure 1B). The serum ratio of L rats was higher at 199 positions, and the serum peak ratio of O rats was higher at 123 positions. There was no difference in peak ratio between L and O rats at 28 positions.

### 3.3. Targeted Metabolomics Data

A targeted metabolomics approach provided identification of some peaks and measurement of the concentrations of these metabolites in serum of L and O group rats (Table 1). We observed significantly higher levels of methionine, cysteine, tryptophan, kynurenic acid, and cysteine/cystine ratio in the L group compared to the O group, and only significantly higher levels of cystine and tyrosine in the O group compared to the L group. The tryptophan/kynurenic acid ratio was slightly higher in lean animals but statistically similar between both groups.

## 4. Discussion

Human obesity is a very complex and puzzling pathological condition where every piece of the puzzle has its own spot and must be placed correctly. Animal modeling is one piece of the puzzle. Adequate and conclusive rat models have a good potential for providing new insight into the problem or for reevaluating existing evidence on the causes and development of obesity [32]. Having a more evident record of metabolic status for both basic physiological (control) and experimentally modified conditions in the Zucker rat model helped us to make more efficient and targeted data interpretation and integration, allowing a better understanding of model outcomes.

Despite the existing published reports regarding metabolic status in Zucker rats [33,34], an untargeted metabolomics approach was never explored, in spite of its potential. Our data that averaged untargeted metabolomics analyses of serum from Zucker rats show that lean (L) and obese (O) rats have substantially different metabolic imprints for the 50 metabolites selected for analysis. Obesity was capable of significantly changing the metabolic profile of 62% of the selected metabolites. These differences were distributed randomly, except for metabolites numbers 44 through 50. In lean rats, metabolite levels in this segment of metabolome were consistently higher compared to obese rats and could be considered excellent candidates for the identification of biomarkers. Additionally, metabolites identified and presented in Table 1 did not belong to this aforementioned cluster.

Untargeted metabolomics analysis of the individual L/O pairs expressed as a peak ratio aimed to express more individual animal variations in the metabolic profiles [35]. More than half of the lean rats (56%) had a higher ratio compared to the obese animals. These data show that the selection of animals for experiments must be carefully considered based on initial metabolic profiles, and data interpretation must be carefully presented.

Targeted metabolomics analysis was performed on several important biochemical pathways. The sulfur-containing essential amino acid methionine plays an important role in biological systems, especially in protein synthesis and biological methylation. In this study, obese rats developed a lower concentration of methionine in serum than what was reported by other investigators using other animal models [36] or from human studies [37]. A deficiency of methionine is capable of significantly modifying multiple extremely important biomethylation reactions and affecting a variety of metabolic pathways [38,39]. Methionine is the source of another amino acid, cysteine, which is a rate-limiting compound in the synthesis of glutathione, the most abundant antioxidant in the body. As a result of methionine deficiency, obese rats developed lower serum concentrations of free cysteine in serum and higher concentrations of cystine, the oxidized, inactive form of cysteine. Cysteine/cystine ratio is an important indicator of oxidative stress [40], and a significantly lower cysteine/cysteine ratio is an important indicator of higher oxidative stress in obese rats. Methionine pathway metabolites involved in biomethylation and regulating level of oxidative stress capable to modulate metabolism of other biochemical pathways involved in development of obesity.

Involvement of the amino acid tryptophan and its metabolites in obesity is supported by other investigators [41]. Lower serum concentrations of tryptophan and kynurenic acid in obese rats in our experimental model supported previously published human data [42,43], making it imperative to study the numerous metabolites of tryptophan and their role in the development of obesity using a metabolomics analytical platform. More than a 25% increase in the serum concentration of the amino acid tyrosine in the obese group of rats needs to be further studied and interpreted as well [44].

In summary, these results represent metabolic snapshot in lean and obese Zucker rats. Our results emphasizing the value and importance of individual metabolic profiles for other investigators to consider these factors for planning and designing stage of experiment. This is a novel approach may explain the individual variation outcome between the rats.

## Figures and Tables

**Figure 1 biomedicines-10-00153-f001:**
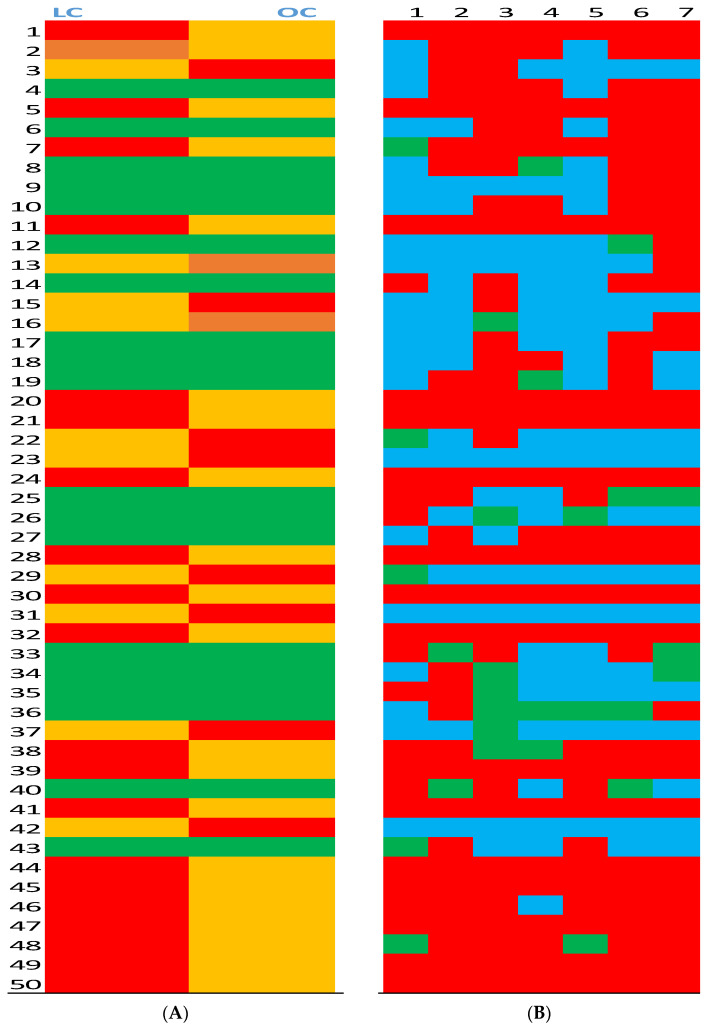
Average peak heights in serum from lean (L) and obese (O) Zucker rats (Panel **A**) and serum peak height ratios in lean (L) and obese (O) Zucker rats (Panel **B**). (Panel **A**): Average height of the corresponding 50 peaks was compared between L and O groups. 

 Significantly (*p* > 0.05) higher average peak intensity in the group. 
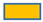
 Significantly (*p* > 0.05) lower average peak intensity in the group. 

 No difference between groups. 
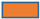
 Marginal (0.1 < *p* > 0.05) difference between groups. (Panel **B**): Pairs of 50 peaks intensity ratio between L and O groups. 

 Peak ratio at least 50% higher in the L group. 

 Peak ratio at least 50% higher in the O group. 
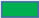
 No difference between groups.

**Table 1 biomedicines-10-00153-t001:** Serum metabolite concentration of lean and obese Zucker rats.

	L	O	*p*-Value
**Methionine,** *nmol/mL*	46.5 ± 4.36	36.4 ± 5.72	0.002
**Free Cysteine,** *nmol/mL*	19.4 ± 4.45	10.1 ± 1.34	0.0001
**Cystine,** *nmol/mL*	8.6 ± 1.05	14.7 ± 2.71	0.0001
**Free Cysteine/Cystine**	2.25 ± 0.434	0.71 ± 0.168	0.0001
**Tryptophan,** *nmol/mL*	63.1 ± 12.18	39.1 ± 9.14	0.001
**Kynurenic acid,** *nmol/mL*	2.26 ± 0.165	1.66 ± 0.332	0.001
**Tryptophan/Kynurenic acid**	27.8 ± 5.61	24.6 ± 8.02	0.2
**Tyrosine,** *nmol/mL*	45.8 ± 8.85	62.2 ± 10.83	0.005

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
