# Peer review of "Metabolic Status of Lean and Obese Zucker Rats Based on Untargeted and Targeted Metabolomics Analysis of Serum"

_biomedicines, 2022, doi:10.3390/biomedicines10010153_

Round 1

Reviewer 1 Report

Melnyk and Hakkak analyzed the serum metabolic profile of female Zucker rats. For untargeted metabolomics, they identified 400 different peaks and selected the 50 most intense for a comparison. For targeted metabolomics, they selected few metabolites and compared their serum concentration between lean and obese rats.

- The experimental analysis has been properly conducted, but the results needs to be improved. The final message does not look so scientifically sound. In the paper of Jonsson et al (A metabolome-wide characterization of the diabetic phenotype in ZDF rats and its reversal by pioglitazone, PLOSone 2018) the authors have analyzed the serum metabolomics of female Zucker rats, going much deeper in the analysis, also in the part related to targeted metabolomics. Since it is not very clear (and it should be better indicated in the materials and methods section), I suppose that in the paper of Melnyk and Hakkak only obese (and not ZDF) rats have been used. Nevertheless, as it is, the analysis seems too poor and incomplete.

 - The authors should try to increase the analysis of targeted metabolites, specifying why they have been selected, their role, pathway of relevance, identifying them as peaks (they should be within the 50 selected peaks). Probably they could try to represent them graphically with medians and IQ range (dot plots) or compared them as ratio/fold increase or decrease (so that they could be more related to what reported in Figure 2).

- Probably, it should be better to regroup figure 1 and 2 together, as different panels. The maps should stay in just one page in order to give a clearer and immediate vision of the data presented and of the final message. If possible, a representative chromatogram could be of help.

- It is important to characterize a little bit better the animal model, especially because the main focus is metabolomics in an obese model. It should be better to indicate something more of the animal included in the study (such as glycaemia and/or serum lipid levels..). A sentence explaining why females have been selected in respect of males should be added. Gender can have an important effect on metabolic profile.

Author Response

Comments and Suggestions for Authors:

Melnyk and Hakkak analyzed the serum metabolic profile of female Zucker rats. For untargeted metabolomics, they identified 400 different peaks and selected the 50 most intense for a comparison. For targeted metabolomics, they selected few metabolites and compared their serum concentration between lean and obese rats.

Reviewer #1:

1- The experimental analysis has been properly conducted, but the results needs to be improved. The final message does not look so scientifically sound. In the paper of Jonsson et al (A metabolome-wide characterization of the diabetic phenotype in ZDF rats and its reversal by pioglitazone, PLOSone 2018) the authors have analyzed the serum metabolomics of female Zucker rats, going much deeper in the analysis, also in the part related to targeted metabolomics. Since it is not very clear (and it should be better indicated in the materials and methods section), I suppose that in the paper of Melnyk and Hakkak only obese (and not ZDF) rats have been used. Nevertheless, as it is, the analysis seems too poor and incomplete.

We thanks this reviewer for the valuable comments.

Answer: We used lean and obese (fa/fa) Zucker rats in this experiment, which are different than ZDF rats. ZDF rats are used as model for diabetic model unlike obese (fa/fa) that we used are for obesity model. 

2- The authors should try to increase the analysis of targeted metabolites, specifying why they have been selected, their role, pathway of relevance, identifying them as peaks (they should be within the 50 selected peaks). Probably they could try to represent them graphically with medians and IQ range (dot plots) or compared them as ratio/fold increase or decrease (so that they could be more related to what reported in Figure 2).

 Answer:

Since this is a short report, in this manuscript, we are showing only initial evidences of involvement three different amino acids in development of obesity in lean and obese zucker rats. We will continue this work on identification of other metabolites within 50 peaks selected in future experiments. Of course, for future study, it will be very interesting to explore possible interaction between these three pathways and establish their serum biomarkers.  

3- Probably, it should be better to regroup figure 1 and 2 together, as different panels. The maps should stay in just one page in order to give a clearer and immediate vision of the data presented and of the final message. If possible, a representative chromatogram could be of help.

Answer:

We combined two figures (1 and 2) on one page as Figure 1 with panel A and B as was suggested.

4- It is important to characterize a little bit better the animal model, especially because the main focus is metabolomics in an obese model. It should be better to indicate something more of the animal included in the study (such as glycaemia and/or serum lipid levels..). A sentence explaining why females have been selected in respect of males should be added. Gender can have an important effect on metabolic profile.

Answer: We expanded this section and addedAnimals homozygous for the fa allele are notably obese by 3 to 5 weeks of age and by 14 weeks of age, their body composition is more than 40% lipid (20). A number of investigators have used this model to study the development, etiology, associated pathogenesis, possible treatment, and putative mechanisms of severe obesity (21). Obese Zucker rats develop hyperinsulinemia and insulin resistance prior to the development of obesity-associated, non-insulin-dependent diabetes mellitus in a manner similar to that in humans (22). Lean Zucker rats, by contrast, exhibit normal metabolic function and are considered ideal controls (23).

 Also, We added the following sentence to method section “we have shown that both obese male and female rats will develop obesity and liver steatosis at the same rates and that there is no difference on between both sexes.

Reviewer 2 Report

This submission compared serum metabolites between 7 lean (L) and 7 obese (O) female Zucker rats to investigate the individual serum metabolic profile. Please conduct the following concerns.

  1. Zucker rats used female strain needs to introduce in clear.
  2. Aims for understanding of the specificity of this model and for widening the base in manipulations seem better to compare with another model of obesity.
  3. Obesity was capable of significantly changing the metabolic profile of 62% of the selected metabolites. Is it same as human subjects? Please discuss it in detail.
  4. Untargeted metabolomics analysis of the individual L/O pairs as an accepted standard test in metabolomics must follow the reference(s).
  5. A deficiency of methionine seems important which must link to obesity in clear.
  6. Numerous metabolites of tryptophan and their role in the development of obesity failed to follow from the indicated two references. Please discuss it in detail.
  7. Conclusion did not show the useful indicator(s) in clear.
  8. Novelty of current study is important.

Author Response

Reviewer #2:

Comments and Suggestions for Authors

This submission compared serum metabolites between 7 lean (L) and 7 obese (O) female Zucker rats to investigate the individual serum metabolic profile. Please conduct the following concerns.

We thanks this reviewer for the valuable comments

1- Zucker rats used female strain needs to introduce in clear.

Answer: We have shown that both obese male and female rats will develop obesity and liver steatosis at the same rates and that there is no difference on between both sexes and also, we added a segment to characterize a little bit better the animal model as other reviewer suggested (24,25).

We expanded the introduction section and addedAnimals homozygous for the fa allele are notably obese by 3 to 5 weeks of age and by 14 weeks of age, their body composition is more than 40% lipid (20). A number of investigators have used this model to study the development, etiology, associated pathogenesis, possible treatment, and putative mechanisms of severe obesity (21). Obese Zucker rats develop hyperinsulinemia and insulin resistance prior to the development of obesity-associated, non-insulin-dependent diabetes mellitus in a manner similar to that in humans (22). Lean Zucker rats, by contrast, exhibit normal metabolic function and are considered ideal controls (23).

2- Aims for understanding of the specificity of this model and for widening the base in manipulations seem better to compare with another model of obesity.

Answer: Well, this is a great suggestion but we have worked with obese zucker rat model for past 15 years and not any other models such as high fat induced obesity in normal Sprague Dawley rats. Of course, in future, we will keep this suggestion in mind.

3- Obesity was capable of significantly changing the metabolic profile of 62% of the selected metabolites. Is it same as human subjects? Please discuss it in detail.

Answer:

We don’t know and it will be great idea and could be very interesting to compere obese Zucker rat model data with future obese human study. The Zucker rat is a model for obesity, which can be different than human obesity (diet and genes). 

4- Untargeted metabolomics analysis of the individual L/O pairs as an accepted standard test in metabolomics must follow the reference(s).

Answer:

We added the new reference (33) below to the discussion of the manuscript:

Ilya Gertsman, Bruce A. Barshop. Promises and Pitfalls of Untargeted Metabolomics. J Inherit Metab Dis. 2018 May; 41(3): 355–366.

5- A deficiency of methionine seems important which must link to obesity in clear.

Answer:

We agree and we added the following sentences to discussion section of the manuscript “Methionine pathway metabolites involved in biomethylation and regulating level of oxidative stress capable to modulate metabolism of other biochemical pathways involved in development of obesity”. 

6- Numerous metabolites of tryptophan and their role in the development of obesity failed to follow from the indicated two references. Please discuss it in detail.

Answer:

In this manuscript, we are trying to show importance and potential in future experiment of analysis tryptophan biochemical pathway numerous metabolites.

7- Conclusion did not show the useful indicator(s) in clear.

Answer:

We changed the summary to read:

In summary, these results represent metabolic snapshot in lean and obese Zucker rats. Our results emphasizing the value and importance of individual metabolic profiles for other investigators to consider these factors for planning and designing stage of experiment. This is a novel approach may explain the individual variation outcome between the rats.

 Novelty of current study is important. We added to summary.

Reviewer 3 Report

The article aims to characterize the metabolomic profile of animal models commonly used to study obesity. It is an interesting topic that can help researcher to gain more information to interpret results in the studies using animal models. I think that the article is publishable in Biomedicines but I have some minor comments/suggestions:

  • For LC-MS detection the authors performed the analysis only in positive ionization mode. Is there any reason for that? Because there are some metabolites that can be detected only in the negative mode. If there is any reason for that it should be stated in the manuscript.
  • In Figure 1 I would appreciate to label the columns and the files of the chart (the files for panel A and the columns for panel B). It would also be nice to have the p-values from the comparatives in the chart or maybe represent the colours in the chart with a gradient using p-values (ex. more intense red for more significant p-values and light red for less significant p-values).
  • Are LC-MS data available in any repository? I would recommend to have them in a public repository for the readers.

Author Response

The article aims to characterize the metabolomic profile of animal models commonly used to study obesity. It is an interesting topic that can help researcher to gain more information to interpret results in the studies using animal models. I think that the article is publishable in Biomedicines but I have some minor comments/suggestions:

For LC-MS detection the authors performed the analysis only in positive ionization mode. Is there any reason for that? Because there are some metabolites that can be detected only in the negative mode. If there is any reason for that it should be stated in the manuscript.

Answer:

We agree with reviewer and we deleted “ in positive ionization mode” from the text. Reviewer is right about negative ionization mode potential in detection of additional metabolites and we will use this mode in our future studies. The main reason why we used positive ionization mode only in present manuscript is that we were focused on metabolic pathways of amino acids methionine and tryptophan and their metabolites were better detectable in positive mode.

In Figure 1; I would appreciate to label the columns and the files of the chart (the files for panel A and the columns for panel B). It would also be nice to have the p-values from the comparatives in the chart or maybe represent the colors in the chart with a gradient using p-values (ex. more intense red for more significant p-values and light red for less significant p-values).

Answer:

We agree with this reviewer about traditional data presentation in untargeted metabolomics commonly accepted.   In this manuscript, we were aiming visually simplify the picture and be concentrated on major differences between groups or individual metabolic profile of rats.

Are LC-MS data available in any repository? I would recommend to have them in a public repository for the readers.

Answer:

This is very good suggestion. We did not present this data in a public repository yet. However; this is only the first step of our study finding and as we obtain more targeted metabolomics analysis in future, we will add them to the public repository.   

Round 2

Reviewer 1 Report

This manuscript has been presented as an article, not as a short report, and for this reason I would expect that the authors added at least something to what I suggested. For example, within the many comments, I suggested to change or try different ways to present the data, but I observe no improvements in this sense.

Also, when I have asked about the animal model, the authors should have reported clinical parameters of their rats, not references about what has already known in literature. This aspect is poor.

As it is presented, the manuscript has not significantly changed compared to the previous version.

Author Response

This manuscript has been presented as an article, not as a short report, and for this reason I would expect that the authors added at least something to what I suggested. For example, within the many comments, I suggested to change or try different ways to present the data, but I observe no improvements in this sense.

Answer: We thanks this reviewer for comments. Originally, we submitted this manuscript as short report since had only 2 figures and one table at that time but now, based on the 1st reviewer’s comment, we combined the 2 figures to one figure and one table only. The editorial office suggested that since this Journal does not have short review then the manuscript is considered as full manuscript and not short review. 

Also, when I have asked about the animal model, the authors should have reported clinical parameters of their rats, not references about what has already known in literature. This aspect is poor.

Answer: We expanded and added the following to introduction:

For past several years, we used this model to investigate the effects of both short and long-term soy diet on liver steatosis. We reported that obesity increases liver steatosis and short and long-term soy protein diet can protect liver steatosis (24, 25). In addition, we have used the obese zucker rat model to study the effects of obesity on breast cancer development and reported that obesity increases mammary tumor development (26-27).

Reviewer 2 Report

It has been improved following the suggestions.

Author Response

2nd Reviewer:

We thanks 2nd reviewer for all of the accepted comments.

Round 3

Reviewer 1 Report

no comments